# Apolipoprotein B and Cardiovascular Disease: Biomarker and Potential Therapeutic Target

**DOI:** 10.3390/metabo11100690

**Published:** 2021-10-08

**Authors:** Jennifer Behbodikhah, Saba Ahmed, Ailin Elyasi, Lora J. Kasselman, Joshua De Leon, Amy D. Glass, Allison B. Reiss

**Affiliations:** Department of Medicine and Biomedical Research Institute, NYU Long Island School of Medicine, NYU Langone Hospital—Long Island, Mineola, NY 11501, USA; Jennifer.Behbodikhah@nyulangone.org (J.B.); Saba.Ahmed@NYULangone.org (S.A.); ailinelyasi1@gmail.com (A.E.); Lora.Kasselman@NYULangone.org (L.J.K.); Joshua.DeLeon@NYULangone.org (J.D.L.); ADG237@optonline.net (A.D.G.)

**Keywords:** apolipoprotein B, LDL-C, atherosclerosis, biomarker, statin

## Abstract

Apolipoprotein (apo) B, the critical structural protein of the atherogenic lipoproteins, has two major isoforms: apoB48 and apoB100. ApoB48 is found in chylomicrons and chylomicron remnants with one apoB48 molecule per chylomicron particle. Similarly, a single apoB100 molecule is contained per particle of very-low-density lipoprotein (VLDL), intermediate density lipoprotein, LDL and lipoprotein(a). This unique one apoB per particle ratio makes plasma apoB concentration a direct measure of the number of circulating atherogenic lipoproteins. ApoB levels indicate the atherogenic particle concentration independent of the particle cholesterol content, which is variable. While LDL, the major cholesterol-carrying serum lipoprotein, is the primary therapeutic target for management and prevention of atherosclerotic cardiovascular disease, there is strong evidence that apoB is a more accurate indicator of cardiovascular risk than either total cholesterol or LDL cholesterol. This review examines multiple aspects of apoB structure and function, with a focus on the controversy over use of apoB as a therapeutic target in clinical practice. Ongoing coronary artery disease residual risk, despite lipid-lowering treatment, has left patients and clinicians with unsatisfactory options for monitoring cardiovascular health. At the present time, the substitution of apoB for LDL-C in cardiovascular disease prevention guidelines has been deemed unjustified, but discussions continue.

## 1. Introduction

Cardiovascular disease (CVD) is the leading cause of death worldwide and its prevalence is expected to continue to rise over the next 15 years [1,2]. According to the American Heart Association (AHA), one in three people will be affected by some form of CVD during their lifetime [3]. The two most common clinical manifestations of CVD are coronary artery disease (CAD) and ischemic stroke [4]. As a major cause of morbidity and mortality in the United States of America and worldwide, complications of atherosclerosis, including myocardial infarction, chronic kidney disease and stroke, are major contributors to the financial burden of healthcare costs [5,6]. The overall approach to reducing CVD morbidity and mortality is focused on primary and secondary prevention and control of modifiable risk factors [7,8]. Despite all efforts, substantial residual risk remains and new lines of attack against atherosclerotic CVD are needed [9]. One avenue that merits exploration is apolipoprotein B (apoB) and its prominent position as a causal factor in atherosclerosis [10].

Atherosclerosis is a progressive disease of large- and medium-sized muscular arteries, characterized by elevated lesions called fibrous plaques that encroach upon the vessel lumen and disturb blood flow. Atherosclerosis is the major cause of CVD. A hallmark of atherosclerosis is the retention of cholesterol-rich low-density lipoprotein (LDL) and other apoB-containing lipoproteins within the arterial wall (Figure 1) [11]. Development of the fatty streak and subsequent transition to fibrous plaque is primarily dependent upon the absorption of modified forms of cholesterol by subendothelial macrophages in an inflammatory setting. Thus, elevated levels of cholesterol in the circulation promote atherosclerosis and CVD [12,13]. Measurement of serum apoB reflects total LDL-C, intermediate density lipoproteins (IDL-C), VLDL-C, and lipoprotein(a) (Lp(a)) particle concentrations because each particle contains exactly one molecule of apoB100. Thus, apoB can be considered a powerful tool for assessment of atherogenic lipid status.

ApoB is a key structural protein component of all major atherogenic lipoproteins. It plays multiple roles in regulating lipid metabolism and is considered to be a physiologically relevant measure of actual number of atherogenic lipid particles. This review will explore the importance of apoB in the atherosclerotic process and its potential role as a biomarker and treatment target. The structure and function of apoB will be discussed. Areas of conflict and controversy will be addressed. In addition, we will focus on future directions of lipid-lowering agents and whether or not apoB should play a more central role in controlling and monitoring dyslipidemia.

## 2. ApoB: Characteristics and Composition

The protein constituents of lipoproteins, apolipoproteins, are found bound to the lipoprotein surface and are largely responsible for the lipoprotein properties, transport and metabolism [13,14,15]. Of the numerous apolipoproteins, apoB is an essential component of VLDLs and its metabolites IDLs and LDLs as well as chylomicrons and their remnants [14]. The apoB particle serves as a frame and is crucial in the maintenance of the structural stability of the lipoprotein (Figure 2) [16,17].

ApoB is encoded by the APOB gene, and occurs in two forms: full-length apoB100, consisting of 4536 amino acids; and apoB48, a truncated form consisting of the N-terminal 2152 amino acids [18,19,20,21]. Although encoded by the same gene, they play distinct roles in physiology. In humans, apoB48 is primarily synthesized and expressed within the intestine and is present in chylomicrons and their remnants [22,23]. In contrast, apoB100 is mainly synthesized and expressed in the liver and is an integral component of VLDL, IDL and LDL (Figure 3) [24,25]. Therefore, of the two forms, apoB100 is more clinically relevant in determining the level of circulating atherogenic lipoproteins [26]. Amongst the plasma apolipoproteins, apoB100 is unique not only because of its large size but also due to its moderate hydrophobicity and inability to transfer between lipoproteins [27,28,29,30].

The apoB100 polypeptide is made up of five domains: βα1, β1, α2, β2, and α3, with α representing a predominantly α-helical structure and β representing a predominantly β-sheet structure [31,32,33]. The N-terminal sequence is vital in the formation of VLDL due to its interaction with the microsomal triglyceride transfer protein (MTP) [34,35]. MTP in the endoplasmic reticulum is required for the first step in apoB generation, transfer of triglycerides, phospholipids and cholesterol esters to the apoB particle [36]. The β-sheet domains are fundamental in establishing irreversible strong bonds to the lipid core that keep lipoproteins anchored to the original apoB particle to which they have been attached [14]. An amphipathic α-helix domain is located between the two β-sheet domains, typical in other apolipoprotein structures as well [37]. Elongation of the β-sheet domains around the lipoprotein imbues apoB100 with a distinct amphipathic quality which allows the stable binding of lipids, especially those within the lipoprotein core [16,32]. These lipid-associating regions are key features for the integrity of LDL particles.

Ribosomes on the surface of the endoplasmic reticulum synthesize apoB100, which is subsequently translocated through a channel to the lumen of the endoplasmic reticulum. ApoB100 secretion is regulated primarily at the post-translational stage [13,38,39]. Typically, secretory proteins are synthesized on the cytosolic surface of the ER followed by rapid translocation through the membrane to the lumen of the ER. However, apoB100 is unlike other secretory proteins in that it becomes associated with the ER membrane very early in the post-translational period, resulting in exposure of the nascent polypeptide to the cytosol [13]. This exposure allows for between 50 and 80% degradation of the newly synthesized apoB100 by hepatocytes. Thus, this rapid co-translational degradation predominantly determines the amount of protein secreted by the cells. Furthermore, whether nascent apoB100 is degraded or secreted depends on the availability of the major lipoprotein lipids, triglycerides, cholesteryl esters, and phospholipids [40]. If the amount of lipid available to lipidate apoB in the endoplasmic reticulum is inadequate or if MTP is not functioning properly, the chaperone protein binding immunoglobulin protein (BiP) will bind to apoB and target it for proteasomal degradation [41,42,43]. Proteasome-independent post-translational degradation of apoB has also been documented [44]. MTP and the amount of lipids available are major determinants of apoB100 translocation in addition to its assembly and secretion [14,45,46]. MTP and apoB100 interact physically at the site of apoB100 translocation across the ER where MTP facilitates the coordinated transfer of lipids and folding of the polypeptide as it exits the ribosome and enters the ER lumen [13,32]. MTP inhibition can block the secretion of apoB100 [13,47]. Subsequent maturation of apoB100 occurs in the golgi apparatus prior to secretion from the hepatocyte [48,49]. The addition of a major load of triglycerides occurs in the golgi apparatus and plays a significant role in determining the size of the VLDL secreted [14,50,51]. Therefore, the amount of triglyceride available in the hepatocytes directly impacts VLDL assembly. States of excess triglyceride production, such as obesity, untreated diabetes mellitus, or in persons consuming a diet high in simple carbohydrates, lead to the formation of triglyceride-rich VLDL [52,53,54]. After a high-fat meal, the concentration of triglyceride-rich large VLDL and chylomicrons increases [55]. The size and density of apoB-containing particles is directly dependent on the availability of triglyceride [56]. A number of additional factors, such as the availability of insulin and fatty acids, can also influence the secretion of these apoB-containing lipoproteins.

In humans, the liver is predominantly responsible for the uptake and disposal of the majority of circulating apoB-containing lipoproteins. Uptake is carried out via three primary receptors: the LDL receptor, heparin sulfate proteoglycans, and scavenger receptor class B type I (SR-BI). The LDL receptor has a half-life of about 25 h and is responsible for the clearance of more than two-thirds of normal LDL, accomplished via binding to a specific site within the α3 domain of apoB100 [16,37,57,58]. The primary binding region that interacts with the LDL receptor, termed site B, is located at residues 3356–3368 of apoB [16,57]. Binding between the LDL receptor and apoB100 only occurs after the polypeptide has undergone a conformational change which results due to the lipolysis of VLDL to LDL [59,60]. Loss-of-function mutations in the LDL receptor or apoB can result in familial hypercholesterolemia, which is characterized by extremely elevated plasma LDL levels, thus leading to accelerated atherosclerosis [61]. Additionally, gain-of-function mutations in PCSK9, a proprotein convertase that accelerates the degradation of the LDL receptor, also results in high LDL concentrations [62]. Following endocytosis, LDL separates from the LDL receptor. The LDL, with a long half-life of 2- to 3-days, is transported to the lysosomes where it is degraded and its lipid cargo released, while the majority of LDL receptors are recycled to the cell surface [63,64,65]. In this way, apoB100 is essential in the catabolism of VLDL, LDL and IDL via its interaction with LDL receptors in the plasma.

In addition to LDL receptor binding sites, apoB100 possesses at least eight potential proteoglycan (PG)-binding sites [66,67]. Of these, two sites have been proposed to act cooperatively in the association with proteoglycans: site A at residues 3148–3158 and site B at residues 3359–3369 [68,69]. The main sites that interact with proteoglycans are sites A and B, sites that also bind to the LDL receptor [14,70]. ApoB-containing lipoproteins bind to PGs via ionic interactions between the negative charged sulfate and carboxyl groups of the glycosaminoglycans (GAGs) and the positively charged basic amino acid lysine and arginine residues of the apoB100 [16,71]. This binding of apoB100 to proteoglycans in the arterial wall is particularly significant as it is considered to be the primary mechanism for the retention of LDL in the subendothelium [72]. The retention and aggregation of apoB-lipoproteins within the arterial wall can be attributed to the intrinsic tendency of damaged or modified apoB100 to aggregate. Proteolysis of apoB by enzymes in the arterial intima can change particle conformation at a molecular level, promoting fusion, aggregation and accumulation in the arterial wall [71,73].

On the other hand, apoB48 is primarily found on chylomicrons and their remnants and is primarily cleared by the heparin sulfate proteoglycan (HSPG) pathway since they do not contain an LDL-receptor binding domain [74,75]. The significance is largely found in diabetes, where high glucose interferes with perlecan biosynthesis, resulting in a decrease in HSPG. This leads to elevated plasma levels of apoB48 containing lipoproteins and ultimately severe postprandial dyslipidemia [76]. While chylomicrons themselves are too large to penetrate the arterial wall, their remnants may do so and are therefore thought to contribute to lipid accumulation in atherosclerotic plaque [77].

## 3. Biomarkers for CVD: LDL-C and ApoB

### 3.1. The Lipid Profile and LDL-C as a Biomarker

Currently, the American Heart Association recommends a screening lipid panel every 4–6 years in patients over the age of 20. However, patients with CVD or at high risk of cardiovascular-related events should be screened more frequently [78,79]. A routine lipid panel consists of total cholesterol, HDL-C, LDL-C and triglycerides, along with cholesterol ratios. Total cholesterol, HDL-C and triglycerides are directly measured, whereas the LDL-C levels are estimated using the Friedewald equation. The Friedewald equation, which is subject to inaccuracy in the presence of high triglycerides and other conditions like diabetes, estimates LDL-C as total cholesterol minus HDL-C and very-low-density cholesterol (VLDL-C), with VLDL-C estimated as triglycerides divided by a fixed factor of 5 [80,81,82,83]. Ultracentrifugation followed by β-quantification is the gold standard to directly measure LDL-C, but this method is impractical, expensive and generally reserved for research use [84,85].

Traditionally, LDL cholesterol has been used to assess the risk associated with CVD and is a frequently used surrogate CVD risk marker in clinical trials [86,87,88,89]. However, LDL-C is an imperfect predictor and many individuals with normal LDL-C levels develop CVD [90].

LDL is generally characterized as a combination of cholesterol contained in a variety of lipoproteins defined by a density between 1.006 and 1.063 g/mL. However, this includes IDL and VLDL, LDL remnants. This range can be further limited to 1.019 to 1.063 g/mL, which is inclusive of LDL only and can be further subdivided for LDL analysis [91]. LDL particles vary in size, composition and density. They have an average diameter of 18–25 nm with roughly 3000 lipid molecules in total. Each LDL particle contains one apo B-100 molecule. LDL cholesterol plays a central role in atherogenesis and estimating risk, but LDL cholesterol content does not reflect LDL particle concentration because metabolic processes involving lipids affect lipid size and composition. The relative ratio of cholesterol to triglycerides in LDL can vary greatly [92]. In a study of 118 healthy individuals, the ratio ranged from 1.8 to 11.5 [93]. This significant amount of variability further suggests that LDL-C alone is not sufficient as an indicator of cardiovascular health, since the particle content differs within individual LDL molecules and risk calculators assume a constant cholesterol concentration per molecule of LDL.

Moreover, in individuals with diabetes and metabolic syndrome, although LDL-C levels are normal, the overall lipid profile is pro-atherogenic with high triglycerides and low HDL-C. An added atherogenic factor in those with diabetes and metabolic syndrome is a significant increase in small dense LDL particles. These unique lipid abnormalities pose an increased risk for cardiovascular events, but the normal LDL-C levels can mislead clinicians, who then may not initiate lipid-lowering therapy [92]. A new biomarker may more accurately represent CVD risk and improved management in these patients.

As of 2013, the ASCVD risk calculator has been recommended to predict the 10-year risk of “hard” cardiovascular events, including nonfatal MI, fatal CAD, nonfatal and fatal stroke [94]. The calculator takes into account age, sex, race, blood pressure, total cholesterol, HDL-C, LDL-C, history of diabetes, smoking, hypertensive treatment, aspirin and 3-hydroxy-3-methylglutaryl-coenzyme A reductase (HMGCoA reductase) inhibitors (statins). Based on the score, high- or low-intensity statins are recommended, along with therapy targeting LDL-C [95].

Current guidelines suggest lowering LDL-C as much as possible, as stated by the American Heart Association and the American College of Cardiology [8]. Analysis of data from the Treating to New Targets (TNT) study, a clinical trial in which stable CAD patients with LDL-C above 130 were randomized to 10 mg or 80 mg of atorvastatin per day for about 5 years, has shown that the predictive power of LDL-C is less significant than that of other potential biomarkers such as apoB and non-HDL-C [96]. However, these levels are still not generally suggested as a first-line target for medical therapies.

For the past 30 years, studies have shown that decreasing LDL-C significantly decreases the risk of coronary heart disease (fatal or non-fatal myocardial infarction) [97]. However, CVD risk reduction achieved via lipid-lowering therapy in most clinical studies does not exceed 30% [87,98]. Furthermore, recent meta-analyses have shown that despite achieving target LDL-C levels with lipid-lowering treatment, there is still a high residual risk of coronary artery disease-related events that should be addressed by clinicians [99,100]. In the PROVE-IT TIMI 22 trial, 22.7% of patients still had major cardiovascular events (MCVE) at 2 years of follow-up, despite achieving recommended LDL-C levels, suggesting that a more sensitive biomarker may be necessary to minimize subsequent cardiovascular events [101].

The 2018 AHA/ACA guidelines place emphasis on using maximum-intensity statins to decrease LDL-C as much as possible, as it will further decrease the risk of cardiovascular events [78]. Aggressive treatment is inarguably beneficial, but estimated LDL-C using the Friedewald equation underestimates true LDL-C levels, and at low levels of LDL-C, the other variables used in the Freidelwald equation are no longer negligible, making the equation inaccurate and LDL-C levels an unreliable estimate of risk [102].

Additionally, even when achieving recommended LDL-C levels, there is still a high residual risk of cardiovascular-related events [99]. Furthermore, in a 2003 survey by the National Cholesterol Education Program, 62% of coronary artery disease patients achieved the LDL-C goal of <100 mg/DL, but only 33% achieved both the LDL-C and non-HDL-C goals [103]. Combined with the studies showing high risk of subsequent MCVE despite achieving target LDL-C levels, this further supports the theory that LDL cholesterol is an incomplete reflection of MCVE risk and new biomarkers may be beneficial in improving our ability to identify persons at risk for CVD and in improving outcomes for CVD patients.

### 3.2. Non-HDL-C as a Biomarker

A number of clinical studies have shown that as triglycerides rise, especially above 400 mg/dL, the Friedewald equation underestimates the true LDL-C value, interfering with assessment of true cardiovascular risk [79,104,105]. Furthermore, nonfasting specimens may be enriched in chylomicrons that contain triglycerides, and this also leads to an underestimation of LDL-C [106].

Recent studies have shown that both apoB and non-HDL cholesterol levels may be equivalent, if not better, indicators of CVD and risk of MCVE compared to LDL-C [107,108,109]. However, the use of non-HDL-C levels has only been advocated as a first-line target by one association, the National Lipid Association, and apoB levels are only recommended as an alternative to LDL-C by the Canadian Cardiovascular Society [110,111]. The 2019 European Society of Cardiology/European Atherosclerosis Society (ESC/EAS) Guidelines for the management of dyslipidemias support the evaluation and consideration of non-HDL-C and apoB as secondary targets for lipid control [112].

Non-HDL cholesterol is highly correlated with apoB levels but is not always consistent. There is a large variability in apoB levels relative to non-HDL-C. Non-HDL-C is a sum of the cholesterol in atherogenic particles, while apoB is found as a single molecule in each atherogenic particle [113].

ApoB was found to be a better predictor than non-HDL-C in identifying more patients with a compromised cardiovascular profile, according to a population-based sample [114]. In addition, a number of differences between non-HDL-C and apoB were noted, such as that increasing levels of non-HDL-C were not associated with a significant increase in the presence of CVD in women. However, an increase in CVD prevalence was noted in both sexes with increasing apoB levels. Moreover, the discriminatory power for the presence of CVD was significantly higher for apoB than for non-HDL-C.

A recent review comparing non-HDL-C and apoB has noted that apoB and non-HDL-C are more accurate measures in ASCVD risk assessment, especially in hypertriglyceridemic individuals, non-fasting individuals, and in those with very low LDL-C concentrations. However, the review did not have enough information to compare apoB and non-HDL-C to each other [115].

A more recent discordance analysis of non-HDL-C versus apoB showed that apoB is the more accurate marker of cardiovascular risk, as apoB can identify elevated numbers of small cholesterol-depleted LDL particles that are neither identified by LDL-C or non-HDL-C. In addition, apoB is better as a target in patients with mild to moderate hypertriglyceridemia (175–880 mg/dL), diabetes, obesity or metabolic syndrome [116].

Overall, little research has been completed to compare apoB and non-HDL-C as predictors of CVD prevalence. From the little research already carried out, it seems that apoB has some advantages, but more work is needed.

### 3.3. ApoB as a Biomarker

Approximately half of all patients with recurrent coronary syndrome have normal cholesterol levels on standard lipid profiles, and despite having achieved the recommended LDL-C levels, these patients are still at high risk of cardiovascular-related events [117,118,119]. At the forefront of promising biomarkers lie apoB and non-HDL-C [117,120].

A single molecule of apoB is present in every atherogenic particle; therefore, it has been proposed as a better predictor of cardiovascular events. Standard LDL-C, on the other hand, is a measurement of lipid concentration in lipoprotein particles that are heterogeneous and vary in size, density and lipid content [121]. Over 90% of total apoB is normally found in LDL particles [122,123]. However, since the lipid composition differs between LDL particles, these values do not strongly correlate with LDL cholesterol levels [113,124,125]. Recent studies have shown that apoB has a higher sensitivity and specificity than LDL-C in predicting cardiovascular events, such as myocardial infarction (MI) in both men and women, independent of age [126]. In a population of Japanese patients with established stable CAD documented by coronary artery stenosis exceeding 75% on coronary angiography, a virtual-histology intravascular ultrasound of the culprit lesions demonstrated greater lesion length and higher plaque volume and percentage of necrotic core volume in patients with high plasma apoB levels when compared to patients with low plasma apoB levels. No correlation was found between apoA1 and the percentage of necrotic core volume of the target coronary artery lesion. In this population, the apoB level was a very good indicator of the size of necrotic core and a potential biomarker for unstable plaque with an advantage over LDL-C [127].

Statins, first-line agents for lipid-lowering, bring about a significantly greater decrease in LDL-C than in apoB levels. This discordance suggests a need for a more precise method of routine lipid monitoring [113].

Additionally, in individuals with LDL-C levels below the median value, apoB can be used to assess MCVE risk, independent of whether or not the atherogenic particles are predominantly LDL-C [16]. This feature is of great significance, especially in diabetic patients, where atherogenicity has a higher level of dependence on lipoproteins other than LDL, such as triglycerides [128]. The inclusivity associated with apoB is one possible explanation for its enhanced predictive capability in determining MCVE risk. ApoB includes LDL-C, VLDL-C, IDL-C and lipoprotein(a), as opposed to LDL cholesterol alone. These other highly atherogenic particles play a key role in CVD and should be accounted for when assessing the risk of subsequent cardiovascular events [129,130,131,132]. Furthermore, apoB is a direct measure of the atherogenic particle number rather than cholesterol concentration, which can vary from one atherogenic particle to another as a result of lipid metabolism [113].

Major drawbacks to any transition to standard measuring of apoB include impracticality and cost, in addition to hesitancy by clinicians to welcome this change. Nevertheless, monitoring lipoprotein levels aside from LDL-C may be critical in managing risk and minimizing morbidity and mortality due to CVD in specific subsets of patients such as those with diabetes, as discussed in Section 4 (How Do Pro-Atherosclerotic Risk Factors Affect ApoB levels?) [99]. In addition, without directly measuring apoB levels, atherogenic dyslipoproteinemias, such as remnant lipoprotein disorder, will continue to be underdiagnosed and undertreated [133].

Measuring apoB by immunoassay may be expensive and time-consuming, and its accuracy may vary [134]. As an alternative, circulating apoB is often estimated using an algorithm, but these values are only approximations based on lipid variables such as the total cholesterol and HDL [135] or LDL and triglycerides [136], and their clinical relevance has not been confirmed.

Interestingly, de Vries et al. have shown that binding of apoB-containing lipoproteins to circulating erythrocytes, detected by flow cytometry, is associated with lower cardiovascular mortality. This inverse relationship between atherosclerosis and levels of erythrocyte-apoB binding can be considered as another means to use apoB as a biomarker [137].

### 3.4. ApoB to ApoA1 Ratio as a Biomarker

While ApoB acts as a major transporter for all atherogenic particles, apoA1 is an anti-atherogenic lipoprotein responsible for transporting cholesterol within HDL-C [138,139,140]. ApoA1 activates lecithin-cholesterol acyltransferase (LCAT), an enzyme that esterifies plasma cholesterol and increases the cholesterol carrying capacity of HDL (Figure 3). The ratio of apoB/apoA1 reflects the balance between the two opposing forces and has been proven to be an accurate indicator of CVD risk. The greater the ratio, the more cholesterol is circulating in the plasma and being deposited into the arterial wall [141].

Multiple prospective studies, including the INTERHEART study and Apolipoprotein-related Mortality RISK (AMORIS) trial, have shown a positive linear association between any risk of a cardiovascular event and increasing apoB/apoA1 ratio [126,142,143,144]. In the AMORIS study, subjects with a first cardiovascular event prior to age 50 exhibited elevated total cholesterol, triglycerides, LDL-C, glucose, apoB, and apoB/apoA-1 ratio compared to controls up to 20 years before the cardiac event [145]. This supports treating and remedying the discrepancy in lipoprotein levels between persons with a high risk for a cardiovascular event at a young age and those without this risk.

Sierra-Johnson et al. used data from The Third National Health and Nutrition Examination Survey (NHANES III) combined with National Death Index information and laboratory analysis of lipids/apolipoproteins in a prospective study of a representative multi-ethnic sample in the United States of America to evaluate cardiovascular risk predictors [146]. When comparing the accuracy of the prediction of coronary heart disease mortality based on apolipoprotein levels, they found that both apoB and the apoB/apoA1 ratio proved better than traditional cholesterol markers. When testing the superiority of apoB/apoA1 over apoB alone, they found that the difference was not statistically significant when adjusting for risk factors, such as obesity, smoking, dyslipidemia, hypertension, diabetes, and high C-reactive protein (CRP). Therefore, there was no proven benefit of using apoB/apoA1 over apoB alone. They thus concluded that apoB is the primary underlying source of the high predictive power of the apoB/apoA1 ratio and the lesser importance of apoA1 in the assessment of risk. On the other hand, the use of apoB/apoA1 has other benefits. The apolipoprotein ratio is associated with insulin resistance in non-diabetic subjects and can therefore be helpful in patients whose phenotypes are independent of apoB [147].

### 3.5. When Does ApoB Show an Advantage over HDL-C?

A number of studies have shown that under specific circumstances, apoB alone is a better predictor of MCVE than the apoB/apoA1 ratio. Data from the INTERHEART study, a global study of over 12,000 cases of first acute myocardial infarction and over 14,000 age- and sex-matched controls, showed that the power of apoB to predict myocardial infarction was consistently higher than non-HDL-C or LDL-C up to age 70. An analysis of patients enrolled in single-center, cross-sectional community-based studies at the University of Pennsylvania found that in Caucasians with type 2 diabetes, plasma apoB, but not LDL-C, may be an indicator of coronary artery calcification beyond traditional risk factors [148]. The apoB/apoA1 ratio was better than the non-HDL-C/HDL-C ratio in predicting the presence of carotid plaques in patients on peritoneal dialysis [149]. The apoB/apoA1 ratio also showed improved accuracy over TC/HDL-C in predicting adverse cardiovascular events in a prospective study of patients with established coronary heart disease (CHD) [144].

In summary, both apoB and apoB/apoA1 are better than traditional cholesterol risk markers for CVD under some circumstances. The use of apoB or apoB/apoA1 holds promise for both evaluating risk and targeting treatment; however, further research is needed to establish whether the ratio adds value over apoB alone [150].

## 4. How Do Pro-Atherosclerotic Risk Factors Affect ApoB Levels?

### 4.1. ApoB, CVD and Demographics

Cardiovascular risk prediction is of high importance for clinicians and patients as a way to assess the risk of developing future symptomatic CVD, thereby allowing for preventive interventions to be instituted in those patients who are most likely to benefit. Accurate estimates of CVD risk are sought in hopes of decreasing CVD morbidity and mortality [151,152,153]. CVD risk is determined by an accumulation of non-modifiable and potentially modifiable factors throughout a lifetime [153,154,155,156,157,158]. Often, multiple risks factors and comorbidities are present in an individual, and the progression of CVD is accelerated by their interaction, not only additively but also synergistically [159,160].

ApoB levels generally do not differ across ethnicities, but the amount of oxidized phospholipid carried per apoB particle is higher in African Americans than in Caucasians or Hispanics, possibly indicating more atherogenic particles in African Americans [161,162]. ApoB levels are lower on average in females than males, and catabolism occurs at a higher rate in females [163]. In a Swedish study, the mean apoB concentration was 1.31 ± 0.35 g/L in males versus 1.22 ± 0.36 g/L in females [164]. ApoB levels are higher in women after menopause, probably because of a slower catabolic rate [165].

### 4.2. Specific Risk Factors: Body Weight, Hypertension, Diabetes

In general, a number of known CVD risk factors have a stronger correlation to apoB levels than to LDL-C and other biomarkers. Patients with normal levels of non-HDL-cholesterol, but high levels of apoB have a higher BMI, waist circumference, systolic blood pressure, fasting insulin and C-reactive protein—all important established risk factors for CVD [159].

A large anonymized clinical laboratory analysis of over 30,000 men and women not on lipid-lowering medications showed that a change in body weight was associated with significant changes in apoB and non-HDL cholesterol that provided information beyond triglyceride and LDL-C measures [166]. A small study of 44 normal weight and 39 obese subjects, all with high LDL-C, comparing dietary intervention with 7 days of either a diet enriched in polyunsaturated fats or saturated fatty acids, showed that the polyunsaturated fat diet lowered both LDL-C and apoB and the BMI × diet interaction showed less improvement in LDL-C and apoB in the obese subjects [167]. A recent meta-analysis with combined data on 335 overweight and obese healthy or hyperlipidemic subjects on calorie-restricted diets found significant reductions in apoB concentrations with 6–12% weight loss. In non-weight loss dietary comparisons, the Mediterranean diet was most closely associated with lower plasma apoB [168].

The accuracy of atherogenic risk attributable to apoB varies at different ages. Although the risk for a coronary event increases with age, the cardiovascular risk associated with the apoB level decreases with age, indicating that apoB may be a better biomarker for CVD in younger patients than in older ones. Results of the CARDIA (Coronary Artery Risk Development in Young Adults) study showed that in persons followed for approximately 25 years beginning as young adults (mean age of 25 years), high apoB at a young age was strongly associated with coronary calcifications, a surrogate CVD risk marker, in middle age (mean age 50 years) [169]. Multiple other biomarkers of cardiovascular risk are also noted to be more accurate in younger patients than in older ones. These include total cholesterol, LDL-C, non-HDL-C, HDL-C and apoA1 [170].

ApoB may be useful in predicting CVD risk in states of insulin resistance such as metabolic syndrome and type 2 diabetes [148,171,172]. ApoB is highly associated with type 2 diabetes and research has shown that apoB levels may be a risk factor for type 2 diabetes [173,174]. ApoB quantification is especially useful in patients with insulin resistance and elevated triglyceride levels above 150 mg/dL, where the Friedewald equation becomes inaccurate and loses its ability to provide a measure of atherogenic load [80,81,82,83,175,176].

In non-diabetic patients, apoB—but not LDL-C—corresponded positively with dysfunctional white adipose tissue and delayed clearance of fat, hyperinsulinemia, insulin resistance, and activation of the IL-1β system, all of which are known risk factors for type 2 diabetes [168,177]. In fact, epidemiological studies have shown that apoB predicts the development of type 2 diabetes as much as 3–10 years in advance of clinical onset [178,179,180].

### 4.3. The Clinical Significance of Small Dense LDL

Small dense LDL particles are associated with excessive CVD risk [181,182]. Among the properties of these small dense LDL particles that make them atherogenic are their ease of penetration into the arterial wall, poor binding to LDL receptors and prolonged circulation time in the bloodstream [183]. Small dense LDLs have enhanced susceptibility to damage from both oxidation and glycation and their level in plasma is highly correlated with plasma triglyceride and apoB concentrations [184,185,186].

Plasma levels of small dense LDL particles may be elevated in patients with inflammatory diseases such as rheumatoid arthritis and systemic lupus erythematosus, which also carry increased CVD risk [187,188]. They are also elevated in states of hyperglycemia such as gestational diabetes, type 2 diabetes and diabetic retinopathy [189,190,191,192]. Viktorinova and colleagues found that the level of small dense LDL in persons with diabetes can be estimated based on the ratio of LDL/apoB such that there is an inverse relationship between these parameters [193]. Since LDL and apoB are more routinely measured than small dense LDL, calculating this ratio can be applied as a CVD risk measure reflecting small dense LDL prevalence [194]. Zheng et al. [195] link compromised clearance of apoB-containing lipoproteins to both hypertriglyceridemia and a shift in LDL composition favoring small dense particles, possibly through a pathway driven by apoC-III. Small dense LDL levels also correlate with the occurrence of peripheral arterial disease [196,197,198].

## 5. ApoB as a Target of CVD Treatment

While apoB may be an applicable target for decreasing risk associated with CVD, only a handful of treatments have been designed to impact the levels of this protein (Table 1). Recent evidence has shown that patients with autoantibodies that directly target apoB have, in fact, lower rates of CVD. Specifically, patients who had plasma levels with a high concentration of antibodies targeting native peptide 210 of apoB-100 had a 45% lower risk of developing a myocardial infarction. This suggests that apoB is not only a prognostic factor but also an important target for minimizing disease progression [199].

Lipoprotein classes constantly exchange their lipid components. One of the most crucial proteins in the lipid metabolism pathway is the cholesteryl ester transfer protein (CETP), a hydrophobic glycated protein that facilitates the exchange of cholesteryl esters and triacylglycerol between HDL and apoB-containing lipoproteins [200,201,202]. Since CETP results in the formation of more apoB-containing triglyceride-rich lipoprotein particles such as VLDL and is inversely associated with HDL particle size and composition, it is considered atherogenic. The inhibition of CETP and cholesteryl ester transfer was expected to increase HDL-C whilst decreasing LDL-C and other particles that harbor apoB, thereby lowering the risk of cardiovascular-related events. However, clinical results were, at first, surprising [203,204]. Three CETP inhibitors, torcetrapib, dalcetrapib and evacetrpib, when tested in humans, which either increased cardiovascular risk or were neutral, while a fourth, anacetrapib, showed modest cardiovascular benefit [205,206,207,208]. Of these drugs, only anacetrapib, in the phase 3 randomized, placebo-controlled REVEAL trial (Randomized Evaluation of the Effects of Anacetrapib through Lipid-Modification), showed a 9% relative risk reduction in major coronary events in persons with pre-existing ASCVD after a median of 4 years of treatment [209,210]. This effect of anacetrapib has been attributed to decreased LDL-C accompanied by a reduction in apoB-containing lipoprotein particles but may also be explained by a decrease in small VLDL particles [211]. Furthermore, anacetrapib may increase the number of cell surface LDL receptors, thus increasing clearance of apoB-containing lipoproteins. Unfortunately, anacetrapib induces excess lipid accumulation in adipose tissue, and so it will not be developed further. Further studies are certainly necessary to refine our knowledge of CETP and improve targeting of this protein in ways that may optimally address cardiovascular risk. A study by Ference et al. in 2019 [212] shed light on the importance of apoB particle number in determining CHD risk. They looked at data on 654,783 people, 91,129 of whom had CHD, and found that triglyceride-lowering variants in the lipoprotein lipase gene and LDL-C–lowering variants in the LDL receptor gene each lessened the cardiovascular event risk to a similar degree when measured based on absolute change in apoB. This suggests that it is the net absolute change in the number of apoB-containing lipoprotein particles rather than the cholesterol content of the LDL particle that determines CHD risk. The implication of this result is that lowering LDL-C by reducing the amount of cholesterol in each particle while not decreasing the particle number will not be effective, even though LDL-C in the lipid profile goes down.

An agent widely utilized to regulate abnormalities in the metabolism of plasma lipids and lipoproteins as well as in the treatment of ASCVD is the B-complex vitamin niacin [213]. Niacin is beneficial in reducing triglycerides and apoB-containing lipoproteins, including VLDL and LDL, through two main mechanisms. One way in which niacin regulates circulating triglycerides and VLDL is by decreasing free fatty acid mobilization from adipose tissue stores. In addition, niacin acts as a noncompetitive inhibitor of hepatocyte diacylglycerol acyltransferase–2, a critical enzyme in the synthesis of triglycerides. With less triglyceride availability, intracellular hepatic apoB degradation accelerates, and this leads to decreased secretion of VLDL and LDL particles by the liver. Niacin improves multiple lipid parameters, and its lipid-altering efficacy is comparable to statins; however, it is underutilized due to the adverse experience of flushing [214,215,216,217,218,219]. This adverse reaction is often mitigated via laropiprant, a prostaglandin D2 receptor (DP1) antagonist that reduces niacin-induced flushing [220,221]. It has been combined with extended-release niacin into a fixed-dose tablet that improves key lipid parameters associated with increased CHD risk in patients with primary hypercholesterolemia or mixed dyslipidemia [222,223,224,225].

Among the human monoclonal antibodies that have established efficacy and safety in achieving desirable LDL-C targets are evolocumab and alirocumab [226]. These PCSK9 inhibitors are used to treat primary hyperlipidemia and have been found to reduce myocardial infarction and stroke in persons with established CVD [227,228]. The PCSK9 enzyme enhances degradation of the LDL receptor; therefore, its inhibition allows for greater accumulation of this receptor, enhancing clearance of apoB-containing lipoproteins and reducing the LDL and apoB particle number [229,230]. It has been suggested that if the target LDL-C level is not achieved with the maximum tolerated statin dose, the next adjunctive therapy to add after ezetimibe would be a PCSK9 inhibitor [231,232].

The inhibition of apoB production may be an effective therapeutic against CVD since fewer apoB particles would lead to lowering of LDL-C and VLDL-C [233,234]. In addition, apoB has been shown to play a key role in LDL-C-induced dysfunction of vascular endothelium, leading the way to apoB-targeted therapy for ischemic CVD [235]. One strategy to reduce apoB is through antisense oligonucleotide technology. Mipomersen, a 20-base-pair single-stranded DNA oligonucleotide that binds to the specific mRNA sequence encoding human apoB-100, had great promise in human trials; unfortunately, liver toxicity associated with this therapy led to its discontinuation [236,237].

Lomitapide, a small molecule microsomal triglyceride transfer protein (MTP) inhibitor, exerts its effects in the liver by binding directly to MTP in the endoplasmic reticulum of hepatocytes and enterocytes. MTP is an intracellular protein that catalyzes the transfer of triglycerides onto apoB within the liver in the formation of VLDL [238]. Lomitapide, via its effect on MTP, prevents the synthesis and secretion of VLDL, thereby causing a decrease in the number of secreted apoB-containing lipoproteins [239,240,241]. It is administered orally and is only indicated for the treatment of homozygous familial hypercholesterolemia as an adjunct to a low-fat diet and other lipid-lowering treatments. Similar to mipomersen, lomitapide has many adverse effects, which has limited its use and tolerability. Trials have shown evidence of elevated transaminases and gastrointestinal side effects, leading to early discontinuation of the drug by a large number of patients. Mipomersen is also a CYPP3A4 inhibitor, and its use is therefore limited in patients taking other medications metabolized by this enzyme [242].

In addition, dabigatran, a small molecule oral anticoagulant that binds competitively and selectively to the catalytic site of thrombin and is used for the prevention of ischemic stroke in atrial fibrillation, has surprisingly been found to decrease apoB levels by as much as 7% [243,244]. Although the exact cause of this pleiotropic effect has yet to be elucidated, Joseph et al. suspect it may be due to competing activity of microsomal carboxylesterases. This unexpected lowering of apoB has the potential to explain the success of dabigatran in the reduction of stroke [244,245]. Further research into the mechanism by which this drug decreases apoB levels may support the importance of apoB in the management of hyperlipidemia.

Bempedoic acid is an inhibitor of hepatic ATP citrate lyase, an enzyme that functions upstream of HMG CoA reductase. The inhibition of ATP citrate lyase decreases intracellular cholesterol biosynthesis and results in LDL receptor upregulation on hepatocytes. This increases the liver uptake of LDL particles and reduces circulating LDL-C, non-HDL-C and apoB levels [246,247].

Another therapeutic approach to CVD is the inhibition of angiopoietin-like protein 3 (ANGPTL3), a secretory glycoprotein that reversibly inhibits the catalytic activity of lipoprotein lipase, the rate-limiting enzyme in triglyceride hydrolysis [248]. Evinacumab, a fully human IgG monoclonal antibody against ANGPTL3, enhances VLDL catabolism, thus lowering LDL, VLDL and triglycerides [249,250]. Evinacumab reduces apoB levels by increasing apoB-containing lipoprotein clearance. It was recently approved by the FDA as an adjunct to other LDL-C-lowering therapies for homozygous familial hypercholesterolemia [251].

Several drugs with primary targets other than apoB can lower apoB incidentally. Fibrates, which reduce plasma triglycerides by inhibiting their hepatic synthesis, also reduce apoB levels by 10 to 20% [252,253]. Gemcabene, a lipid-lowering drug in development that works by decreasing apoC-III and thus increasing VLDL clearance, also lowers apoB, LDL-C and CRP [254].

## 6. Future Perspectives

New therapies that lower apoB are on the horizon and, despite the problems with mipomersen, most promising is the application of antisense technology. Gene silencing with antisense oligonucleotides is being used to directly interfere with PCSK9 production. Inclisiran is a synthetic, double-stranded siRNA that yielded a sustained reduction of over 20% in apoB in phase 2 human trials [255].

When currently available lipid-lowering therapy is administered, the maximum decrease in risk of MCVE is 30–40%. However, poor compliance is a major obstacle in many patients [256,257]. This widespread issue can be resolved with a vaccine exerting long-term effects [258,259]. Antigen-specific immunoprotection via vaccination is a recent potential approach to prevention and treatment of chronic diseases [260]. Mechanisms include the production of antibodies, T-cell anergy, and the induction of regulatory T cells [261]. The specificity of apoB peptides is key in avoiding side effects by host defenses. The two forms of atherosclerosis vaccines being developed are antibody-inducing and regulatory T cell-inducing. Vaccines based on apoB-derived peptides have shown promising results by targeting an immune response via regulatory T cells and reducing atherosclerotic lesions in mice [262]. Treating CVD with vaccines will face many challenges but holds a lot of promise. Years of research lie ahead of us in exploring the role vaccination could play in the treatment of CVD, as well as other inflammatory diseases [263].

## 7. Conclusions

Cardiovascular disease remains the leading cause of death worldwide [264]. Recent data have shown a decline in mortality from CHD in the United States of America, but the rate of decline is decelerating and, in younger adults under age 45, there is a lack of progress in reducing cardiovascular deaths [265,266]. In this younger group, a rise in diabetes mellitus and obesity may be hindering improvement in cardiovascular mortality [267]. It has been predicted that by the year 2035, over half of the US population will suffer from some form of cardiovascular disease and projected annual costs may exceed 1 trillion dollars [268]. Pharmacotherapy based on cholesterol management and lipid profile is the cornerstone of treatment and prevention. However, even with lipid-lowering therapy, the absolute risk of cardiovascular-related events remains elevated, and many patients do not achieve lipid goals, most frequently those at high cardiovascular risk [269,270]. Current American guidelines focus on LDL-C-targeted therapy; however, as shown in this review, there is a preponderance of data supporting a role for apoB in cardiovascular risk prediction. ApoB has been proposed as a better predictor of MCVE because a single molecule is found in every atherogenic particle and LDL-C levels alone can miss elevated particle numbers [271].

The evaluation of apoB mass in plasma by mass spectrometry allows the characterization of the proteome of the particles. With mass spectrometry, detection of individual peptide components and comparison of molar ratios may improve risk prediction. Data are supplied on individual subfractions within apoB-containing particles and the extent of oxidation of phospholipids on apoB particles can be determined [192,272,273]. The method is antibody-independent and can be automated. Mass spectrometry was applied in a study by Bodde et al. that found a strong association between plasma levels of apoA1, apoB, and the apoB/apoA1 ratio and first ST-segment elevation myocardial infarction [274].

The wide acceptance of LDL-C coupled with the added expense and complication of measuring apoB has thus far prevented a major shift toward clinical application of plasma apoB at the point-of-care [275,276]. This may change as standardization of apoB measurement improves and as data supporting the benefits of apoB in cardiovascular health assessment accumulate [277,278].

## Figures and Tables

**Figure 1 metabolites-11-00690-f001:**
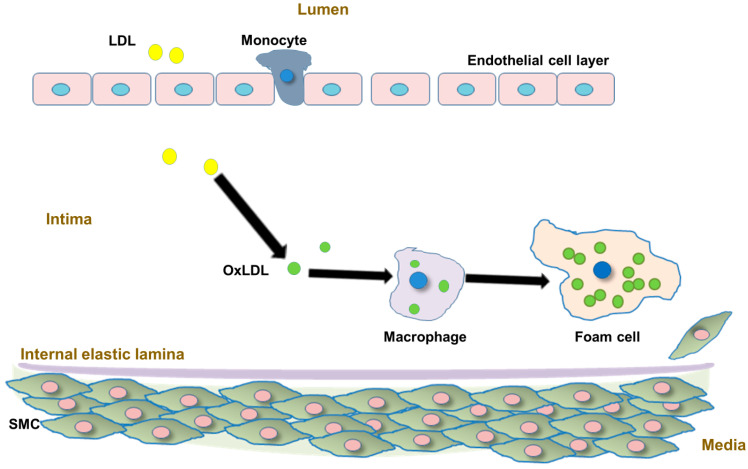
Atherosclerosis involves apoB-containing lipoproteins. The atherosclerotic process begins with compromise of the endothelial barrier, allowing apoB-containing LDL cholesterol to migrate into the arterial intima. Activated endothelium fosters attachment, migration and proliferation of vascular smooth muscle cells (SMC) and macrophages. Retained apoB-containing lipoproteins are oxidatively modified within the vascular intima. Oxidized (ox)LDL contains protein components, creating a net negative charge, making the particles highly attractive to macrophages. Phagocytosis allows for the accumulation of lipids within macrophages, producing foam cells. OxLDL-laden foam cells amass and form the fatty streak and eventually the lumen-narrowing atheromatous plaque that restricts blood flow. Additionally, inflammatory signaling pathways are activated, leading to increased cell migration and LDL modification.

**Figure 2 metabolites-11-00690-f002:**
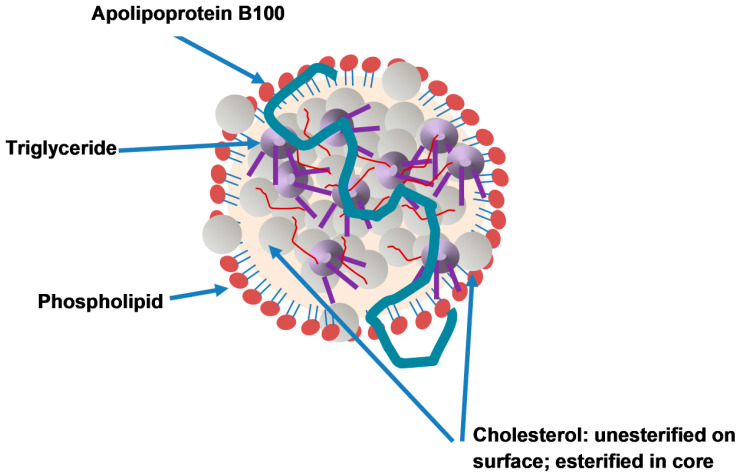
Apo B in a lipoprotein particle. A single molecule of apoB is irreversibly bound to the surface of the lipoprotein particle. The lipid content of the surface consists of a monolayer of phospholipid and free cholesterol. The hydrophobic core is composed of triglyceride and cholesterol esters.

**Figure 3 metabolites-11-00690-f003:**
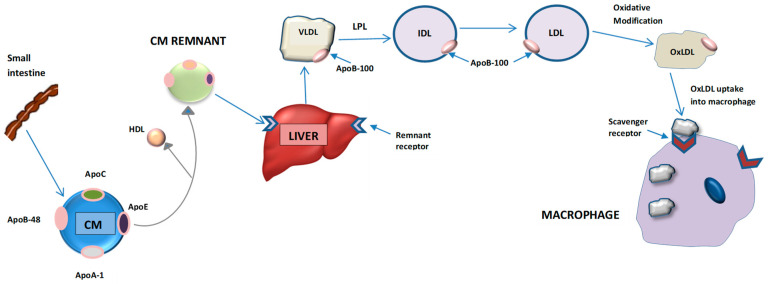
Apolipoprotein (apo) B in atherogenic lipoprotein formation. ApoB is the critical structural protein of all atherogenic lipoproteins. It has two major isoforms: apoB48 and apoB100. ApoB48 is found only in chylomicrons (CM) and chylomicron remnants. It mediates the secretion of chylomicron particles from the intestines. Chylomicron remnants are taken up by the liver. Free fatty acids generated from chylomicron remnants are used by the liver to make triglycerides that are incorporated into nascent VLDL. VLDL particles, each harboring a single apoB100 molecule, are secreted from the liver carrying endogenous, hepatically synthesized triglycerides. VLDL particles shrink with loss of surface components to HDL and are catabolized to IDL by lipoprotein lipase (LPL). Then, IDL is converted to LDL. It is LDL that carries the majority of the circulating cholesterol. LDL can be oxidatively modified and taken up by macrophages which leads to excess accumulation and the formation of foam cells.

**Table 1 metabolites-11-00690-t001:** Lipid-lowering therapies that reduce ApoB.

Therapy	Type of Compound	Mechanism of Effect on ApoB
statins	Competitive inhibitors of HMG-CoA reductase	Lower apoB concentration by decreasing entry of apoB-containing lipoproteins LDL and VLDL into plasma
anacetrapib (development discontinued)	Small molecule oxazolidinone	Potent selective CETP inhibitor. Reduces apoB-containing lipoprotein particles
niacin	Nicotinic acid (vitamin B3)	Modulates liver synthesis of triglycerides, limiting VLDL assembly, resulting in intrahepatic apo B degradation
evolocumab and alirocumab	Fully human anti-PCSK9 monoclonal antibodies	PCSK9 inhibitors increase hepatic LDL receptors, which remove apoB-containing LDL particles from the circulation. Lp(a) also decreased, mechanism not understood
mipomersen (development discontinued)	Synthetic phosphorothioate antisense oligonucleotide apoB inhibitor	Prevents translation of the apoB mRNA into protein, leading to decreased VLDL and LDL
lomitapide	Small molecule that binds directly to and inhibits MTP	Inhibition of MTP in hepatocytes and enterocytes by lomitapide reduces plasma levels of all ApoB-containing lipoproteins.
dabigatran	Novel, synthetic, specific, non-peptide thrombin inhibitor	Antithrombotic effect due to binding competitively to the active site on human thrombin. ApoB lowering is a pleiotropic effect, mechanism unclear.
bempedoic acid	8-hydroxy-2,2,14,14-tetramethylpentadecaned-ioic acid	Inhibits ATP-citrate lyase in the liver, which decreases liver cholesterol synthesis and reduces serum LDL levels by upregulating LDL receptors.
evinacumab	Fully human monoclonal antibody directed against ANGPTL3	Antagonizes ANGPTL3-mediated inhibition of lipoprotein lipase and endothelial lipase, increasing clearance of apoB-containing lipoproteins.
fibrates	Amphipathic carboxylic acids that act as peroxisome proliferator receptor α agonists	Reduce plasma triglycerides by inhibiting their hepatic synthesis and increasing their catabolism. Lower LDL-C, non-HDL-C and apoB.
inclisiran	siRNA conjugated to triantennary N-acetylgalactosamine carbohydrates	Inhibits PCSK9, thereby reducing levels of apoB-containing lipoproteins.

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
