# Peer review of "Apolipoprotein B and Cardiovascular Disease: Biomarker and Potential Therapeutic Target"

_metabolites, 2021, doi:10.3390/metabo11100690_

Round 1

Reviewer 1 Report

Overall, this is very good review for Apolipoprotein B in the context of cardiovascular disease and its potential as biomarker candidate for CVD risk. The author references are up-to-date and relevant to the field. 

Some minor points to improve includes:

  1. Figure 1 lack proper labeling or descriptions. There is no indication of lumen, one of the cell (grey in color) which I suspect is monocyte? is not labelled. Each layer of the vessels should be properly labelled. While the figure legend is very descriptive, the figure itself is oversimplified.
  2. Figure 2 gives me an impression that ApoB is an integral protein rather than surface protein. The weight/thickness of the arrow could be thinner to make the figure looks more refine, or legends can be added instead of arrows to define each components. 
  3. Figure 3 is too crowded with no clear descriptions or message. Figure legend did not provide a clear message as well. I would appreciate if author could provide a clearer descriptions (so many arrows but what or which pathway do they represent?) 
  4. Line 212-They average.... grammatical error
  5. Line 222-229, message is not clear in this paragraph, please rephrase or re-write this paragraph. 
  6. Line 241, what are the other potential biomarkers? 

Author Response

Response to Comments of Reviewer 1:

We thank the reviewer for thoroughly scrutinizing our manuscript. As requested, we have revised the manuscript and addressed the specific comments of the reviewer. The revised sections are delineated in red in a marked copy of the manuscript text.

Below, we provide a point-by-point response to the reviewer’s comments.

COMMENT 1:

  • Figure 1 lack proper labeling or descriptions. There is no indication of lumen, one of the cell (grey in color) which I suspect is monocyte? is not labelled. Each layer of the vessels should be properly labelled. While the figure legend is very descriptive, the figure itself is oversimplified.
  • Figure 2 gives me an impression that ApoB is an integral protein rather than surface protein. The weight/thickness of the arrow could be thinner to make the figure looks more refine, or legends can be added instead of arrows to define each components.
  • Figure 3 is too crowded with no clear descriptions or message. Figure legend did not provide a clear message as well. I would appreciate if author could provide a clearer descriptions (so many arrows but what or which pathway do they represent?)

RESPONSE:  We have made the suggested changes.

  • Figure 1. We have labeled clearly and improved the quality of the figure.
  • Figure 2. We have adjusted the arrows and made it clear that apoB is over the surface of the particle.
  • Figure 3. We have taken out extraneous material to eliminate crowding and rewritten the entire figure legend.

COMMENT 2:

  • Line 212-They average.... grammatical error
  • Line 222-229, message is not clear in this paragraph, please rephrase or re-write this paragraph.
  • Line 241, what are the other potential biomarkers?

RESPONSE:  We have corrected line 212. We have rephrased lines 222-229. We have named other potential biomarkers (apoB and non-HDL-C).

We thank the reviewer and believe that the manuscript is improved as a result of the reviewer’s input. We hope you will agree, and will decide in favor of accepting our report at this time.

Reviewer 2 Report

  • Major point: the term “therapeutic target” in the heading implies that a significant part of the review is addressed to the discussion of data from interventional clinical trials testing ApoB reduction and/or markers of cardiovascular outcomes. Actually this is poorly addressed by authors, who present few (and not recent) data from medium-sized epidemiological studies. Which is the actual relation between cholesterol reduction and ApoB reduction? This could be an interesting topic to discuss about. I would implement a Table or a figure showing mechanisms and data regarding lipid lowering therapies to reduce ApoB.
  • Regarding therapeutic targets, an important discussion might be the comparison between classical small molecules as compared to biotechnological compounds. Which are the perspectives of treatment in the near future?

Author Response

Response to Comments of Reviewer 2:

We thank the reviewer for thoroughly scrutinizing our manuscript. As requested, we have revised the manuscript and addressed the specific comments of the reviewer. The revised sections are delineated in red in a marked copy of the manuscript text.

Below, we provide a point-by-point response to the reviewer’s comments.

COMMENT 1: Major point: the term “therapeutic target” in the heading implies that a significant part of the review is addressed to the discussion of data from interventional clinical trials testing ApoB reduction and/or markers of cardiovascular outcomes. Actually this is poorly addressed by authors, who present few (and not recent) data from medium-sized epidemiological studies. Which is the actual relation between cholesterol reduction and ApoB reduction? This could be an interesting topic to discuss about. I would implement a Table or a figure showing mechanisms and data regarding lipid lowering therapies to reduce ApoB.

RESPONSE: Since we did not mean to imply that the paper would focus exhaustively on therapeutic trials targeting ApoB, we have changed the title to: Apolipoprotein B and Cardiovascular Disease: Biomarker and Potential Therapeutic Target. We have added the table (Table 1) as suggested and included some additional treatments such as inclisiran and bempedoic acid and incorporated a number of new references.

COMMENT 2: Regarding therapeutic targets, an important discussion might be the comparison between classical small molecules as compared to biotechnological compounds. Which are the perspectives of treatment in the near future?

RESPONSE: We have added further discussion of small molecules and put a lot of information in the table. We also added a section entitled “Future Perspectives”.

We thank the reviewer and believe that the manuscript is improved as a result of the reviewer’s input. We hope you will agree, and will decide in favor of accepting our report at this time.

Reviewer 3 Report

Dear Editor,

I really appreciate the work from Behbodikhah et al. The role of apoB has been well addressed by the authors within this review. The images are clear and suitable for publication. I suggest to include a table gathering the impact of different lipid lowering therapies (statins, PCSK9i, bempedoic acid) on apoB. This would increase the value of the paper in clinical practice.

Author Response

Response to Comments of Reviewer 3:

We thank the reviewer for thoroughly scrutinizing our manuscript. As requested, we have revised the manuscript and addressed the specific comments of the reviewer. The revised sections are delineated in red in a marked copy of the manuscript text.

Below, we provide a point-by-point response to the reviewer’s comments.

COMMENT 1: I really appreciate the work from Behbodikhah et al. The role of apoB has been well addressed by the authors within this review. The images are clear and suitable for publication. I suggest to include a table gathering the impact of different lipid lowering therapies (statins, PCSK9i, bempedoic acid) on apoB. This would increase the value of the paper in clinical practice.

RESPONSE: We appreciate this suggestion and have added the table (Table 1) as well as expanded discussion and new references.

We thank the reviewer and believe that the manuscript is improved as a result of the reviewer’s input. We hope you will agree, and will decide in favor of accepting our report at this time.

Round 2

Reviewer 2 Report

1) Page 12/27, lines 504/506: I suggest authors to revise the discussion of reference#203. Actually, the naturally occurring LDL-C cholesterol is not a result of the HDL charge promoted by the lifelong exposure to CETP variants summed in the CETP score. Vice versa, I would better suggest how this is dependent to the LDLR residual activity.

This seminal information has to be discussed in light of the following additional reference: Ference BA et al. JAMA 2019 Pubmed ID: 30694319.

2) Among observational trials studying the ApoB as biomarker for pre-clinical atherosclerosis, I would better discuss on few data looking at the superior relevance of ApoB mass instead of ApoB lipoproteins cholesterol content.

Author Response

Response to reviewer 2

We thank the reviewer for thoroughly scrutinizing our manuscript. As requested, we have revised the manuscript and addressed the specific comments of the reviewer. The revised sections are delineated in red in a marked copy of the manuscript text.

Below, we provide a point-by-point response to the reviewer’s comments.

Comment 1

1) Page 12/27, lines 504/506: I suggest authors to revise the discussion of reference#203. Actually, the naturally occurring LDL-C cholesterol is not a result of the HDL charge promoted by the lifelong exposure to CETP variants summed in the CETP score. Vice versa, I would better suggest how this is dependent to the LDLR residual activity.

This seminal information has to be discussed in light of the following additional reference: Ference BA et al. JAMA 2019 Pubmed ID: 30694319.

Response

We have modified the discussion as requested and added the Ference paper with description of highlights and importance of this paper.

Comment 2

2) Among observational trials studying the ApoB as biomarker for pre-clinical atherosclerosis, I would better discuss on few data looking at the superior relevance of ApoB mass instead of ApoB lipoproteins cholesterol content.

Response

In the Conclusion, we have added a paragraph of discussion with references addressing the issue of apoB mass.
